# Evaluating the quality of life among melasma patients using the MELASQoL scale: A systematic review and meta-analysis

**Yuan Zhu[1], Xiaofang Zeng[2], Jieya Ying[1], Yujia Cai[1], Yu Qiu[2], Wenzhong Xiang[2]\***

**1** Department of Dermatology, Hangzhou Third People's Hospital, Zhejiang Chinese Medical University, Hangzhou, China, **2** Department of Dermatology, Hangzhou Third People's Hospital, Affiliated Hangzhou Dermatology Hospital of Zhejiang University School of Medicine, Hangzhou, China

\* xiangwenzhong@126.com

## Abstract

### Background

According to the literature, pigmentary disorders have a significantly negative impact on a person's health-related quality of life. Moreover, among pigmentary disorders, incidence of melasma ranks high. The Melasma Area and Severity Index (MASI) is the scale that is generally used to evaluate a melasma-affected area and its severity. However, the relationship between the MASI and Melasma Quality of Life (MELASQoL) scores, as well as the impact of melasma on patients' quality of life, remain unclear.

### Objectives

To explore the influence of melasma on patients' lives, analyze the relationship between the MASI and MELASQoL scores, and identify the factors that may be influencing the quality of life of patients with melasma.

### Methods

Two reviewers independently searched four databases (PubMed, Embase, the Cochrane Library, and Web of Science) for literature on quality of life of patients with melasma. In addition to an epidemiological study, a cross-sectional study, and validation studies, gray literature was also included. StataSE version 16 software was used for the meta-analysis. The score of each item on the MELASQoL scale was determined using a random-effects model.

### Results

Fourteen studies with a total of 1398 melasma patients were included in the systematic review, four of which were eligible for meta-analysis. The relationship between the MELAS-QoL and MASI scores was found to be mixed. Five studies concluded that the MASI and MELASQoL scores were statistically correlated, while seven studies found no statistical correlation between the two. It is obvious that melasma causes emotional distress and has a negative impact on patients' social lives. Patients were most bothered by the appearance of

**Data Availability Statement:** All relevant data are within the manuscript and its Supporting Information files.

**Funding:** WX received a research grant (No. 2020364003) from Zhejiang Medical and Health Science and Technology Program, China (URL: http://www.msttp.com/); The study also received a research grant (NO. 81872517) from National Natural Science Foundation of China. The funder had no role in study design, data collection and analysis, decision to publish, or preparation of the manuscript.

**Competing interests:** The authors have declared that no competing interests exist.

their skin condition. However, the MELASQoL score had no definite correlation with patient characteristics such as age, education levels, and history.

## Conclusion

Melasma has a significant negative impact on patients' quality of life. Thus, evaluating the quality of life of patients with melasma should not be ignored. Additionally, utilization of the MELASQoL scale should be considered in the care plan. Further studies with larger sample sizes are needed to confirm the relationship between melasma and quality of life.

## Introduction

Pigmentary disorders refer to changes in skin color due to a decrease or increase in pigment. They include melasma, vitiligo, and freckles. Although most pigmentary disorders do not cause significant harm to physical health, several studies have reported that pigmentary disorders may have a significant negative impact on a person's health-related quality of life [1].

Melasma, a tan pigmentation spot formed on the face, is caused by a variety of factors. It primarily affects women with Fitzpatrick III-VI skin, certain genes, UV exposure, specific hormone levels, or those using drugs or cosmetics with specific ingredients [2–6]. Currently, treatment for melasma remains challenging. In several studies [7, 8], melasma has been shown to rank high among the incidence of pigmentary disorders. Moreover, the immense impact of melasma on patients' quality of life is apparent [9–11]. Therefore, the importance of timely assessment and interventions to improve patients' quality of life is gradually becoming highlighted in the process of diagnosis and treatment. According to the World Health Organization (WHO), quality of life refers to individuals' perceptions of their position in life within their cultural context, value systems, goals, expectations, standards, and concerns [12, 13]. The measurement of quality of life is increasingly becoming a part of the overall assessment of patients' health [14] as it provides a more detailed understanding of a patient's health status [15].

The Melasma Area and Severity Index (MASI) is the scale that is generally used to evaluate the area affected by melasma and its severity. In prior studies, the Dermatology Life Quality Index (DLQI), Skindex (16 and 17-item versions), and mental-health-related anxiety or depression scales have been used to evaluate the impact of dermatological skin diseases on quality of life [4, 15, 16]. In these indices, equal weightage is given to physical and psychological distress, and the particular effects caused by dermatological conditions. However, melasma has a greater impact on the psychosocial aspects, compared to the physiological aspects, of patients' lives [17]. To address this shortcoming and meet the need to assess the quality of life among melasma patients, Balkrishnan et al. developed a new tool in 2003 [17] based on the SKINDEX-16 (7 items) and a questionnaire on changes in skin color (3 items). This tool is called the Melasma Quality of Life (MELASQoL) scale, and it was expected to be more relevant because of its focus on the psychosocial aspects. The new scale was also believed to reflect the severity of the disease. The MELASQoL comprises 10 questions, scored from 1 (not bothered at all) to 7 (bothered all the time); the higher the score, the lower the quality of life. The domains of the MELASQoL include "Work," "Family relationships," "Social life," "Sexual relationships," "Recreation leisure," "Physical health," "Money matters," and "Emotional well-being." These domains have been proven to have high correlations with the SKINDEX16, DLQI, and the skin discoloration evaluation, and they have a high internal consistency (0.95)

[17, 18]. In addition, social life, recreation and leisure, and emotional well-being were found to have the largest effect. Since the MELASQoL is a subjective rating scale that places a greater emphasis on the emotional and psychosocial aspects [19], it was considered to be better, to some extent, at differentiating patients from various groups (patients with emotional, psychiatric, or psychological problems, as well as those who are being actively treated for their melasma, scored higher on average than others). Therefore, it is considered to be better than the other dermatological quality of life indicators [20].

Hitherto, most studies regarding the application of the MELASQoL scale have been conducted in the respective authors' countries and regions or are local epidemiological surveys of melasma patients. There have, to date, been no qualitative or quantitative reviews of the quality of life in patients with melasma. Therefore, what is currently known about the relationship between melasma disease, patient characteristics, past history, and patient's quality of life is not clear. Moreover, the disease's degree of influence on patients needs to be studied. The present study, thus, aimed to explore the impact of melasma on patients' quality of life by analyzing the relationship between the disease's severity and MELASQoL scores and identifying the factors that may be influencing the quality of life of patients with melasma.

## Materials and methods

### Search strategy

In this systematic review and meta-analysis, two reviewers independently searched databases in accordance with the Preferred Reporting Items for Systematic Reviews and Meta-Analyses (PRISMA) guidelines (see S1 Checklist for the complete PRISMA checklist). A review was performed by searching the PubMed, Web of Science, Cochrane Library, and Embase databases for studies published in English using the keywords "(quality OR questionnaire OR scale OR MELASQoL OR epidemiologic) AND (chloasma OR melasma OR melanosis)." The literature search encompassed studies published in English up to February 1, 2021. No starting date restrictions were applied to any of the databases.

### Inclusion and exclusion criteria

For the systematic review:

The inclusion criteria were as follows. (1) For participants: patients visited the local medical clinic and were diagnosed with melasma by a qualified medical practitioner. (2) For study type: observational studies. (3) For publication type: all article types (original article, review, letter, editorial etc.) (4) For outcomes: studies that used the MELASQoL scale to assess the quality of patients' lives. (5) For language: English language papers only.

The exclusion criteria were as follows. (1) For participants: patients who were unwilling or unable to understand the questionnaire. (2) For study type: interventional studies, case reports, and studies that do not report primary data. (3) Incomplete or incorrect data.

For the meta-analysis: Further screening of the studies included in the systematic evaluation, the inclusion criterion was "studies that provided a specific number of people for each item on the MELASQoL scale."

### Data extraction

Two authors independently extracted data on article information (first author, publication year, country, journal, and sample size), demographics of melasma patients (age, gender, education, marriage, and family income), conditions of patients with melasma (duration of illness and MASI score), quality of life assessment (the evaluation scale and scores as well as

influencing factors related to the scale's scores), and other factors that might be related to the quality of life.

For the meta-analysis, the total number of participants and the responses to each domain of the MELASQoL were extracted. In the MELASQoL scale, a 4 indicates "No feelings either way," while a 5 indicates "Sometimes bothered." We set a choice of more than 5 points as causing bother to the patients, and we then added up the scores. StataSE 16 software was used to calculate the rate and standard error rate for each item, and 95% confidence intervals (CI) were calculated as well. A meta-analysis was performed on the individual domains of the MELASQoL instruments, the results of which were reported in at least three previous studies. A random-effects meta-analysis was applied when significant heterogeneity was detected between studies (Cochran's Q $p<0.01$ or $I^2>50\%$); otherwise, the fixed-effects approach was employed.

## Study quality assessment

Based on the Joanna Briggs Institute (JBI) Critical Appraisal Checklist, a nine-item tool was used to evaluate the quality of the studies, including sampling frame, sampling technique, subjects and settings, and data collection tools [21]. The total possible score of the Critical Appraisal Checklist is 18 points, where "Yes" is 2 points, "Not clear" is 1 point, and "No" is not scored. Of which, a final score greater than or equal to 70% is considered high-quality literature. The two reviewers independently evaluated the risk of bias, and disagreements regarding the quality were resolved by a third author.

# Results

## Study selection

The initial search yielded 2,483 references. After eliminating irrelevant content and using the "Endnote" duplicate removal tool, we applied the inclusion and exclusion criteria to the remaining 456 references. Two authors independently reviewed all the abstracts to determine whether the studies were eligible to be included in the systematic review. Of the 19 studies that evaluated the quality of life of patients with melasma, 15 utilized the MELASQoL scale. To increase the data and prevent partial data loss, we also included two letters, two conference abstracts, and one editorial. Of these, one of the conference abstracts did not provide the data, results, and conclusion we needed, therefore, we excluded it. Thus, we were left with a total of fourteen useful studies and other pieces of literature (Fig 1), four of them were included in the meta-analysis.

**Study characteristics.** Of the 14 studies (including nine original articles, two letters, two conference abstracts, and one editorial) that were included in the systematic review, four were also included in the meta-analysis. Table 1 lists the details. Of the 14 studies, six were carried out in Asia (India, Singapore, Indonesia, and Korea), four in South America (Brazil), three in Europe (Turkey, France, and Spain), and one in Oceania (Australia).

Data from a total of 1,398 patients with melasma were included, of which 1,286 (91.99%) were female patients. The average age of the patients ranged from 31 to 56 years; however, other characteristics such as marital status, education, and income could not be fully summarized due to differences in classification and the possibility of data loss.

**Study quality.** We evaluated the 14 articles included in this study. The quality of the studies was evaluated using the JBI cross-sectional quality assessment tool (Table 2). The main risks affecting the quality of the included literature are the participants' sampling strategy, the valid methods for condition identification, response rate, and the management of it. Eight of the included studies had a low risk of bias. Fig 2 shows the quality assessment carried for the included studies.

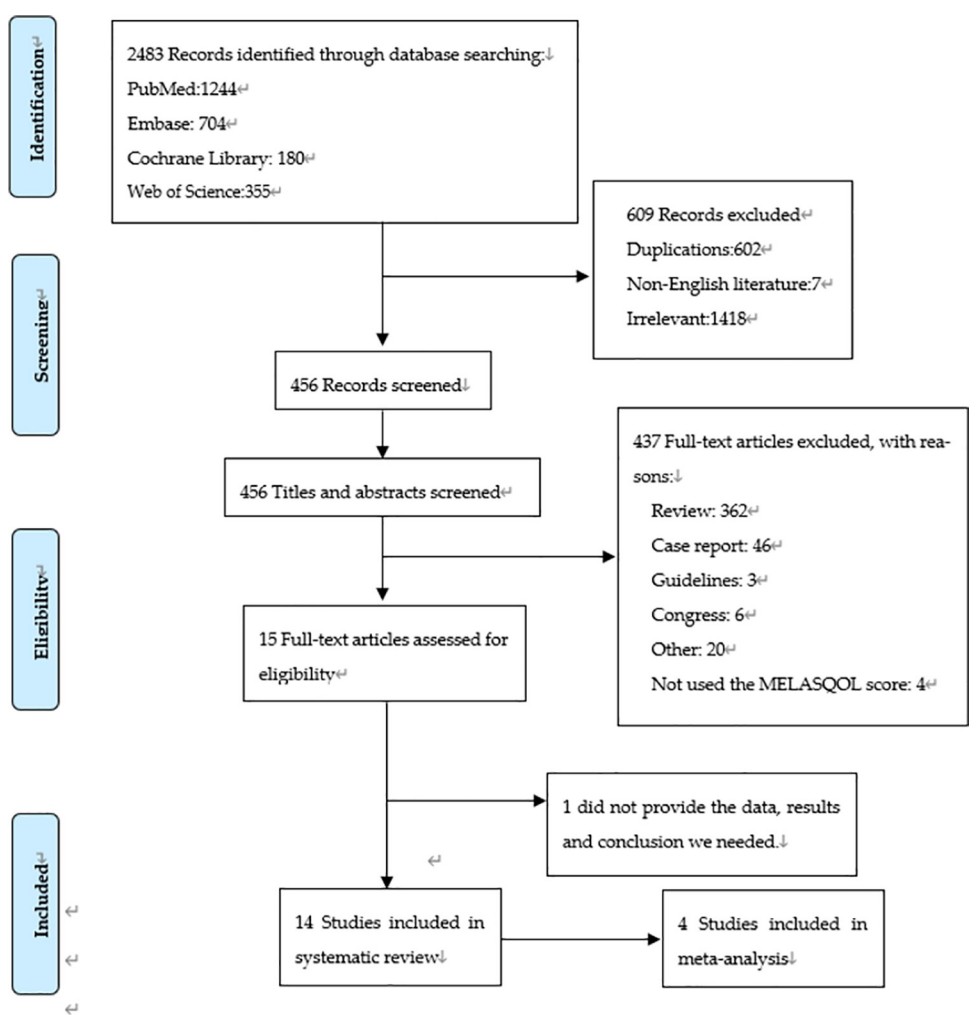

**Fig 1. PRISMA flow diagram.**

## The relationship between MELASQoL and MASI

Generally speaking, the quality of life among patients with melasma should be positively correlated with the severity of melasma. However, this was not what we observed in our review. Twelve studies statistically analyzed the relationship between the MASI and MELASQoL scores. Of which, five concluded that the MASI and MELASQoL scores were statistically correlated [14, 22–25], while seven studies found no statistical correlation between them [2, 19, 20, 26–29]. In terms of correlation, one study concluded that the MELASQoL and MASI scores were strongly correlated ($\rho = 0.809$); the others' results showed that, although the two were correlated, the correlation was not strong ($\rho = 0.233$–$0.350$). Therefore, we concluded that the relationship between the MELASQoL and MASI scores does not have a strong correlation or is at least mixed.

## The influence of melasma on quality of life

Despite the mixed result regarding the relationship between the MELASQoL and MASI scores, the articles in this review explained that the burden due to melasma is influenced by the psychosocial distress that is associated with the pigmentation itself rather than by the formatted

**Table 1. Details on studies included in the systematic review.**

| Publication year | Author | Country | N | Gender | Mean MASI score | Assessment scale | Mean Scale Score | Whether the MASI and MELASQoL scores were related |
|---|---|---|---|---|---|---|---|---|
| 2006 | Dominguez et al. [22] | Spain | 99 | F | 10.89 | MELASQoL-S | 42.49 | YES ($\rho = 0.233$, $p < 0.05$) |
| 2006 | Cestari et al. [12] | Brazil | 300 | 292 F, 8 M | / | MELASQoL-BP | 44.40 ± 14.90 | / |
| 2008 | Freitag et al. [26] | Brazil | 83 | F | 10.60 ± 6.64 | MELASQoL-P | 37.50 ± 15.20 | NO ($\rho = 0.17$, $p = 0.109$) |
| 2009 | Dogramaci et al. [14] | Turkey | 114 | F | / | MELASQoL-Tr | 29.90 ± 14.60 | YES ($\rho = 0.35$, $p < 0.001$) |
| 2010 | Misery et al. [27] | French | 28 | F | / | MELASQoL-F | 20.90 (15.90–25.90) | NO |
| 2013 | Park et al. [23] | Korea | 77 | F | / | MELASQoL | 38.10 ± 16.60 | YES ($p < 0.05$) |
| 2015 | Yalamanchili et al. [30] | India | 140 | 95 F, 45 M | / | MELASQoL | The study did not provide the average value, but it did give specific scores. | / |
| 2015 | Ikino et al. [28] | Brazil | 51 | F | / | MELASQoL-BP | 34.40 ± 13.50 | NO |
| 2016 | Harumi et al. [19] | Singapore | 49 | F | 12.10 ± 6.50 | MELASQoL | 25.60 ± 15.30 | NO |
| 2016 | Sarkar et al. [24] | India | 100 | F | 20.00 ± 7.50 | MELASQoL-Hi | 37.19 ± 18.15 | YES ($\rho = 0.809$, $p < 0.05$) |
| 2018 | Kothari et al. [29] | India | 141 | 105 F, 36 M | 9.07 ± 6.12 | MELASQoL-Hi | 28.61 ± 12.92 | NO ($\rho = 0.151$, $p = 0.074$) |
| 2019 | Anderson et al. [20] | Australia | 31 | 29 F, 2 M | 4.70 ± 3.10 | MELASQoL | 55.00 ± 10.60 | NO ($\rho = 0.033$, $p = 0.09$) |
| 2018 | Pollo et al. [25] | Brazil | 155 | 134 F, 21 M | 8.00 (5.00–14.00) | MELASQoL-BP | 30.00 (17.00–45.00) | YES ($\rho = 0.35$, $p < 0.05$) |
| 2019 | Jusuf et al. [2] | Indonesia | 30 | / | 13.07 ± 4.99 | MELASQoL | 39.97 ±12.07 | NO ($p = 0.797$) |

objective disease severity scales, such as MASI scoring. Furthermore, since the influence of melasma on patients' quality of life is significant, it should not be ignored.

The meta-analysis pointed out [12, 20, 28, 30] that the most disturbing aspects for patients were "The appearance of your skin condition" (0.86, 95% CI 0.75–0.98), "Frustration about your skin condition" (0.81, 95% CI 0.74–0.88), and "Embarrassment about your skin condition" (0.79, 95% CI 0.73–0.86). The least disturbing aspect was "Your skin condition makes it hard to show affection" (0.20, 95% CI 0.10–0.29) (Fig 3). Most authors mentioned that melasma negatively influences emotional well-being, social life, and recreation and leisure, which, in turn, reduces patients' quality of life.

In summary, the influence of melasma on patients' quality of life was mainly reflected through the following aspects.

1. Emotional distress: Patients expressed dissatisfaction, frustration, embarrassment, and depression related to their skin condition. They reported that it made them feel unattractive and expressed that it had affected their social life. This aspect has the most significant influence on their quality of life.

2. Social life: Melasma makes patients feel unattractive to others and reduces their desire to be around people or interact with them. However, melasma does not prevent the patients from showing affection to others. This aspect bothered them sometimes.

## The relationship between MELASQoL scores and patient characteristics

Each study mentioned different patient characteristics or factors influencing the MELASQoL score (listed in Table 3). Four studies found that the MELASQoL score was not associated with

**Table 2. The quality of the included studies.**

| | Was the sample frame appropriate to address the target population? | Were the study participants sampled in an appropriate way? | Was the sample size adequate? | Were the study subjects and the setting described in detail? | Was the data analysis conducted with sufficient coverage of the identified sample? | Were valid methods used for the identification of the condition? | Was the condition measured in a standard and reliable way for all participants? | Was there appropriate statistical analysis? | Was the response rate adequate, and if not, was the low response rate managed appropriately? | Total score |
|---|---|---|---|---|---|---|---|---|---|---|
| Dominguez et al. 2006 | Yes | Yes | Yes | Yes | Yes | Not clear | Yes | Yes | Not clear | 16 |
| Cestari et al. 2006 | Yes | Yes | Yes | Yes | Yes | Not clear | Yes | Yes | Not clear | 16 |
| Freitag et al. 2008 | Yes | Yes | Yes | Not clear | Yes | Not clear | Yes | Yes | Not clear | 15 |
| Dogramaci et al. 2009 | Yes | Not clear | Yes | Yes | Yes | Not clear | Yes | Yes | Not clear | 15 |
| Misery et al. 2010 | Yes | Not clear | No | Yes | No | Not clear | Yes | Yes | Not clear | 11 |
| Park et al. 2013 | Yes | Not clear | No | Yes | No | Not clear | Yes | Yes | Not clear | 11 |
| Yalamanchili et al. 2015 | Yes | Not clear | Yes | Yes | Yes | Not clear | Yes | Yes | Not clear | 15 |
| Ikino et al. 2015 | Yes | No | No | Yes | No | Not clear | Yes | Yes | Not clear | 10 |
| Harumi et al. 2016 | Yes | Not clear | No | Not clear | No | Not clear | Yes | Yes | Not clear | 10 |
| Sarkar et al. 2016 | Yes | Not clear | Yes | Yes | Yes | Not clear | Yes | Yes | Not clear | 15 |
| Kothari et al. 2018 | Yes | Not clear | Yes | Yes | Yes | Not clear | Not clear | Yes | Not clear | 14 |
| Anderson et al. 2019 | Yes | Not clear | No | Yes | No | Not clear | Yes | Yes | Not clear | 11 |
| Pollo et al. 2018 | Yes | Not clear | Yes | Yes | Yes | Not clear | Yes | Yes | Not clear | 15 |
| Jusuf et al. 2019 | Yes | Not clear | No | Yes | No | Not clear | Yes | Yes | Not clear | 11 |

age [14, 19, 20]. Three studies concluded that the course of the disease was not related to the quality of life [19, 20, 28]. Two studies showed that there is no correlation between educational attainment and MELASQoL score [14, 19]. Three studies demonstrated that the MELASQoL scores were higher in patients who had received treatment for melasma [22, 27, 29]. However, there were also different opinions, as detailed in Table 3.

In addition to the aforementioned most-discussed aspects, one study discussed some unusual topics and concluded that patients with a history of mental illness had a lower quality of life [26], this was also seen in patients with polycystic ovarian syndrome [29]. Interestingly, prior studies reported that people with mixed diets (compared to vegetarians) and those who had never used oral contraceptives (compared to oral contraceptive users) had a lower quality of life [29].

In summary, we found that those who had been previously treated for melasma scored higher on the MELASQoL scale, compared to those who had not. Other patient characteristics such as age, course of disease, and educational attainment were the most discussed factors across all the articles, however, none of them showed a consistent correlation with the quality

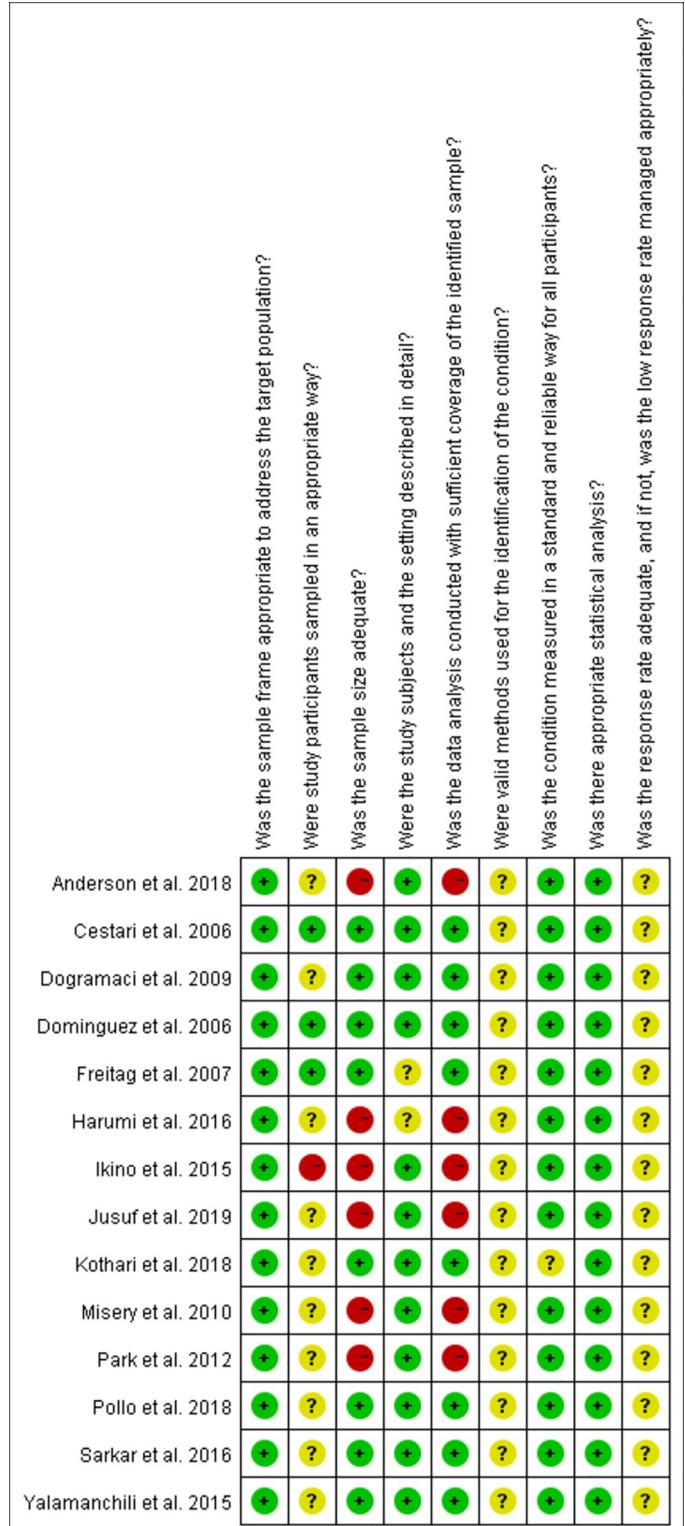

**Fig 2. Study quality assessment of the included studies based on the JBI critical appraisal checklist.**

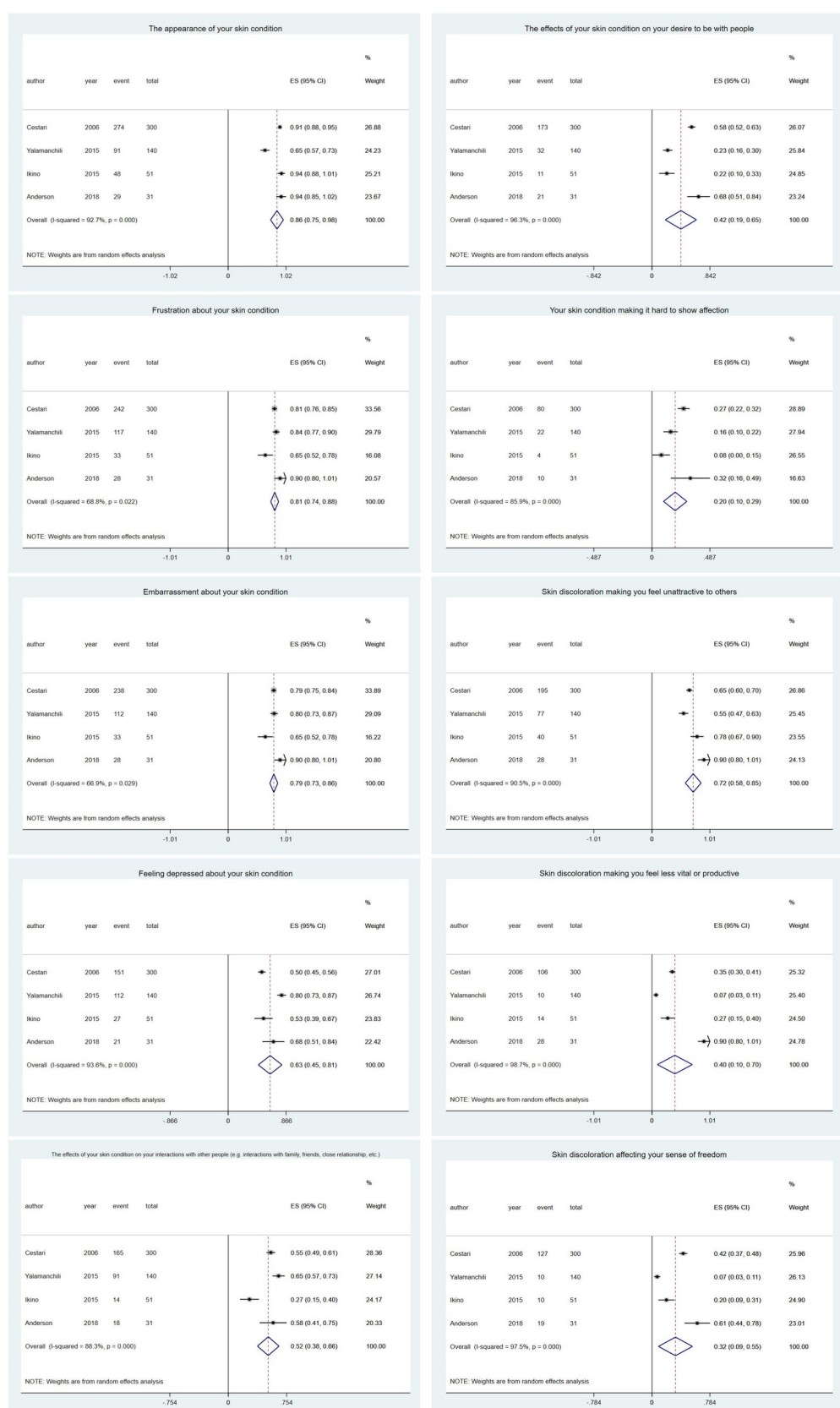

**Fig 3. Meta-analysis of the studies that reported on the individual domains of the MELASQoL scale.**

**Table 3. Controversial factors that influence the quality of life of patients with melasma.**

| | Statistically different from MELASQoL scale scores: | Not statistically different from MELASQoL scale scores: |
|---|---|---|
| Age | 1. Younger patients had higher scores (p = 0.014) (feeling bothered) [28]. | 1. Age and MELASQoL score (p > 0.05) [20]. |
| | 2. Younger patients had higher scores (p = 0.005) (feeling unattractive) [28]. | 2. Age and MELASQoL score (p > 0.05) [19]. |
| | | 3. Age and MELASQoL score |
| | 3. Older patients had higher scores (p = 0.047) [24]. | (p > 0.05) [29]. |
| Previous treatment for melasma | 1. Previously treated had higher scores (47.7 vs. 35.5, p < 0.05) [22]. | / |
| | 2. Previously treated had higher scores (33.9 vs. 27.63, p = 0.036) [29]. | |
| Duration of disease | Shorter duration of disease had higher scores (p = 0.044) [29]. | 1. Duration of disease and MELASQoL score (p > 0.05) [20]. |
| | | 2. Duration of disease and MELASQoL score (p > 0.05) [28]. |
| | | 3. Duration of disease and MELASQoL score (p > 0.05) [19]. |
| Positive family history | / | 1. Positive family history and MELASQoL score (p > 0.05) [28]. |
| | | 2. Positive family history and MELASQoL score (p > 0.05) [29]. |
| Educational level | 1. Those with no formal education had higher scores than patients with at least a seventh-grade education (49.4 vs. 39.3, p < 0.05) [22]. | Education level and MELASQoL score (p > 0.05) [19] |
| | 2. Those with less than eight years of education had higher scores than patients with more education (44 vs. 34.4, p = 0.024) [26]. | |
| Employment | / | 1. Employment and MELASQoL score (p > 0.05) [19]. |
| | | 2. Employment and MELASQoL score (p > 0.05) [29]. |
| History of mental illness | Patients with previous diagnoses of psychiatric diseases had higher scores (42.8 vs. 35.4, p = 0.044) [26]. | / |
| Diagnosed with other diseases | Patients with polycystic ovarian disease had higher scores (44 vs. 27.91) [29] | 1. Thyroid disease and MELASQoL score (p > 0.05) [28]. |
| | | 2. Thyroid disease and MELASQoL score (p > 0.05) [29]. |
| Medication history | Those with no history of oral contraceptive use had higher scores (29.97 vs. 21.23, p = 0.013) [29]. | / |
| Dietary habits | Those that consumed a mixed diet had higher scores than vegetarians (30.15 vs. 24.85, p = 0.027) [29]. | / |

of life of patients with melasma. Regarding other factors, such as mental illness or a history of other diseases, dietary habits, and medication history, it is difficult to have a definitive result due to the small number of articles which examined these factors as well as the small sample sizes of the studies. However, we believe that these factors may be relevant and therefore, further studies are needed to gain a better understanding.

To distinguish differences in the quality of life of patients with melasma across different countries, we compared studies from different regions (Table 4). It can be seen that there is a variation in MELASQoL scores, with the lowest scores reported in France and the highest in Australia. No correlation was found between the MASI and MELASQOL scores—that is, there

**Table 4. MASI and MELASQOL scores in different countries.**

| | Country | MASI score | MELASQoL score | Whether the MASI and MELASQoL scores were related |
|---|---|---|---|---|
| Asia | India | 20.00 ± 7.50 | 37.19 ± 18.15 | YES (ρ = 0.809, p < 0.05) |
| | India | 9.07 ± 6.12 | 28.61 ± 12.92 | No (ρ = 0.151, p = 0.074) |
| | Indonesia | 13.07± 4.99 | 39.97 ± 12.07 | No (p = 0.797) |
| | Korea | / | 38.10 ± 16.60 | Yes (p < 0.05) |
| | Singapore | 12.10 ± 6.50 | 25.60 ± 15.30 | No |
| South America | Brazil | / | 44.40 ± 14.90 | / |
| | Brazil | / | 34.40 ± 13.50 | No |
| | Brazil | 8.00 (5.00–14.00) | 30.00 (17.00–45.00) | Yes (ρ = 0.350, p < 0.05) |
| | Brazil | 10.60 ± 6.64 | 37.50 ± 15.20 | No (ρ = 0.170, p = 0.109) |
| Europe | Spain | 10.89 | 42.49 | Yes (ρ = 0.233, p < 0.05) |
| | Turkey | / | 29.90 ± 14.60 | Yes (ρ = 0.350, p < 0.001) |
| | French | / | 20.90 (15.90–25.90) | No |
| Oceania | Australia | 4.70 ± 3.10 | 55.00 ± 10.60 | No (ρ = 0.033, p = 0.09) |

was no indication that there is a relationship between these two. Interestingly, Australia, which has the highest MELASQoL score in the known data, has the lowest MASI score.

## Discussion

This systematic review and meta-analysis focused on the influence of melasma on patients' quality of life. This influence is subjective; thus, to minimize the errors due to different evaluation scales, we only included studies that used the MELASQoL scale to assess patients' quality of life. We found a mixed correlation between MELASQoL and MASI scores; however, most literature reported that they are unrelated or that the correlation is weak. Since we believe that clinical severity should not be the only criterion used to assess the burden of patients' skin conditions, we focused on the subjective experiences of a group of patients. "Feelings" do not have a clear standard of evaluation since they change according to the context. Even when melasma is not severe, it can cause psychological distress and thereby, affect patients' quality of life.

We found that melasma had the greatest influence on patients' leisure, emotional health, and social lives. Studies showed that melasma causes confusion, frustration, embarrassment, and loss of confidence among patients; moreover, it makes them feel unattractive and affects their relationships.

Patient characteristics that affect the quality of life among people with melasma vary greatly across different regions and populations. Overall, there was no clear correlation between the quality of life and patient characteristics such as age, educational background, or the duration of melasma (the evidence currently available evidence is inconclusive). The only factor that yielded a consistent result was that patients who had previously been treated for melasma reported a lower quality of life compared to those who had never been treated. This may be because melasma seriously affects the quality of life, leading patients to seek medical treatment. Several studies that aimed to explore the treatment of melasma reported a statistically significant decrease in MELASQoL scores after treatment, despite the varied intervention methods [31–36]. Sarkar et al. found that MELASQoL scores decreased (from 47.27 to 37.93) after 12 weeks of sunscreen use, which suggested that sunscreen use subjectively improved the quality of life of patients with melasma [37]. Thus, interventions that effectively treat melasma can improve the quality of life.

This is the first systematic review and meta-analysis that examined the quality of life of patients with melasma, thereby, helping fill the gap in literature. Based on our findings, we

believe that the severity of melasma disease is not a comprehensive indicator of patients' quality of life; at least, the relationship between the MASI and MELASQoL scores is mixed. Melasma affects every aspect of a patient's life, among which emotional stress is the most significant consequence, followed by the negative impact on social life. Finally, existing studies suggest that patients who had previously received treatment reported a lower quality of life, however, no consistent association between patient characteristics and quality of life has been found. Therefore, more in-depth studies are needed to address this issue.

## Strengths and limitations

To date, this study is the most comprehensive review exploring melasma and its impact on quality of life. We examined the included studies, analyzed the relationship between the MELASQoL and MASI scores, and identified different aspects of life that were affected by melasma and patient characteristics that may influence the quality of life. These have not been examined in previous reviews. Those that do exist are typically limited to local conditions. We reviewed the existing relevant literature from various regions and countries to gain a more comprehensive understanding of the quality of life of melasma patients. However, there are some limitations to this study. First, we found that most prior studies only included female participants and therefore, the proportion of male patients was small. Thus, there is little evidence regarding the impact of gender differences on the quality of life. Second, we only included studies that used the MELASQoL scale as the evaluation tool. Further, differences in patients' disease perceptions, treatment practices, presence of co-morbidities, and country- or culture-specific factors influence MELASQoL, and as a result, the scores vary across patient settings. Thus, we concluded that situations differ according to the region and people involved. In addition, due to the limitations related to the data and research methods, we were only able to summarize the results, from prior studies, regarding the relationship between the MASI and MELASQoL scores. This factor is essential and requires further research.

## Recommendations for practice and further research

Although there is no clear evidence regarding the relationship between MELASQoL and MASI scores, the former is a focus during treatment. The effect of melasma on the quality of life cannot be assessed only based on the severity of the disease, as patient characteristics have a significant impact on the quality of life as well. Since the impact of melasma on quality of life is mainly reflected through their mental health, an objective assessment that pays close attention to the needs of patients and psychological comfort is required. For patients with mental-health-related issues, a multidisciplinary team of psychologists and physicians should intervene for a better therapeutic effect. This also reflects the importance of individual differences in the treatment for skin diseases that cause disfiguration. Last, although the MELASQoL scale has been validated in multiple studies, to confirm its clinical applicability it needs to be utilized it clinical settings more and not just in scientific research.

Future research should not be limited to the MELASQoL scale. Different findings might be obtained by expanding the sample size or by simultaneously using other scales, or even new survey methods (such as open-ended questions), to measure the quality of life of patients with melasma. Second, attention must be paid to the influence of the various factors, mentioned in the literature, that may affect the occurrence of melasma, such as demographic background, dietary habits, and mental stress, and how they affect the development of melasma. In addition, we did not find any longitudinal studies that examined at the quality of life in patients with melasma (for example, when patient characteristics change, does the impact of melasma on

their quality of life change accordingly). We believe that observing changes in an individual is more meaningful than observing changes within groups.

Due to the wide utilization of the MELASQoL scale there were problems such as difficulty in semantic interpretation, no prior qualitative analysis of disease perception, and low representation of feelings and self-esteem dimensions [38, 39]. To address these issues, in 2018, a multidimensional tool was developed and verified [38]. Comparing the MELASQoL scale with this new multidimensional tool may provide a more in-depth understanding of the impact of melasma on patients' quality of life.

## Conclusion

Studies on the quality of life of patients with melasma are necessary as they can provide new insights into the patient's experience. Our review confirms the importance of the MELASQoL scale. The findings suggest that when diagnosing and treating melasma, the MELASQoL scale can be helpful in assessing the disease and clarifying its impact on patients' lives, which in turn can help healthcare professionals provide timely comfort and counseling. Moreover, a multi-disciplinary approach to treatment can help achieve better results. Further studies with larger samples are needed to explore the factors influencing the quality of life of melasma patients from different cultural backgrounds and ethnicities.

## Supporting information

**S1 Checklist. PRISMA checklist.** PRISMA statement for reporting systematic reviews and meta-analyses.
(DOC)

**S1 Table. Search strategy.**
(DOCX)

## Author Contributions

**Data curation:** Yuan Zhu, Xiaofang Zeng, Yujia Cai.

**Formal analysis:** Yuan Zhu, Jieya Ying.

**Investigation:** Yuan Zhu.

**Methodology:** Yuan Zhu, Xiaofang Zeng.

**Project administration:** Wenzhong Xiang.

**Resources:** Yuan Zhu, Wenzhong Xiang.

**Supervision:** Yuan Zhu, Yujia Cai, Wenzhong Xiang.

**Writing – original draft:** Yuan Zhu.

**Writing – review & editing:** Jieya Ying, Yu Qiu.

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
