## [Decision Letter · Decision Letter 0]

11 Aug 2021

PONE-D-21-20728

Quality of life among melasma patients: a systematic review using qualitative methods

PLOS ONE

Dear Dr. Xiang,

Thank you for submitting your manuscript to PLOS ONE. After careful consideration, we feel that it has merit but does not fully meet PLOS ONE’s publication criteria as it currently stands. Therefore, we invite you to submit a revised version of the manuscript that addresses the points raised during the review process.

ACADEMIC EDITOR:

While the study is interesting and relevant. The main concerns are regarding the clarity and depth of the results. They seem to be a bit superficial and also there are details missing in the statistics as pointed out by the reviewers. Also, you would need to bring out the knowledge gaps and future avenues for research in more details

We look forward to receiving your revised manuscript.

Kind regards,

Feroze Kaliyadan, M.D.

Academic Editor

PLOS ONE

“NO”

4. Thank you for stating the following in the Funding Section of your manuscript:

“This research was funded by the Zhejiang Medical and Health Science and Technology Program, China (grant number 2020364003)(URL:http://www.msttp.com/); National Natural Science Foundation of China, grant number 81872517. The funder had no role in the study design, data collection and analysis, decision to publish, or preparation of the manuscript”

“NO”

“NO authors have competing interests.”

Reviewers' comments:

Reviewer #1: The authors have addressed an important question to fill the gaps in existing knowledge.

Did the authors consider doing a scoping review as a precursor to a systematic review to map the literature, identify the types of studies available.

Inclusion and exclusion criteria lacks clarity. Please see how the criteria serves the objective of the article- "To explore the influence of melasma on patients’ life, analyze the relationship between the MASI and MELASQOL scores, and summarize the possible factors influencing the quality of life in patients with melasma".

Lines 144- 146 : "Despite the mixed result between the MELASQOL and MASI scores, the articles in this review explained that the burden from melasma is obviously influenced by psychosocial distress associated with the depigmentation itself". Did the authors mean hyperpigmentation?

Though the authors have stated that the included studies are heterogeneous under' limitations', they have not discussed the same in depth from a qualitative point of view.

Reviewer #2: Dear Authors,

You have submitted a systematic review of studies using MELASQOL to study the QoL in melasma.

There is no previous systematic review on this topic. Your findings following the literature review are along the expected lines and are nothing new. However, it becomes clear that there are several lacunae in this field, and I suggest that you should clearly bring out the research gaps in this area, and how to address them in your manuscript.

Please see my more specific comments below:

1. The title should include MELASQOL, since you have excluded other studies on QoL which did not use MELASQOL

2. I am not familiar with the term Gray literature. Please clarify.

3. There are several English language errors. Please address them carefully.

4. Though you mentioned how many points your included studies had, it is not clear what does it mean? Were these good quality studies? What about risk of bias?

5. The data in Table 1 is not clearly visible, particularly the 1st alphabet / figure is blocked by the column line.

6. What statistical test was used to test the correlation between MASI and MELASQOL. You have only provided p-value, that too only as > or <0.05. Please be specific. What about the strength of association, based on the value of correlation coefficient? Only providing p-values is not enough.

7. p10, line 146 - 'depigmentation' Pls check.

8. Please provide a brief review on MELASQOL in the introduction part. Focus on its development and validation. How is it better than general skin instruments such as DLQI and Skindex? What domains of life does MELASQOL cover?

9. Please provide SD and range along with mean values.

10. The authors mention the effect of melasma on QoL as 'significant' and 'enormous' at various places. On what basis? What does a MELASQOL score of 46 mean? Has the interpretability of MELASQOL been studied?

11. You have mentioned physical health of patients was prominently affected? Which item of MELASQOL deals with this aspect? Is it the item 'less vital or productive'? If so, the score for this item is one of the lowest (Fig 2).

12. The term 'personal history' is not clear. Variables such as age, gender, education etc are considered to be patient characteristics.

13. Overall, results are not reported very well and appear superficial. It is not clear how many studies reported on which variable. Further, certain aspects are not very amenable to qualitative analysis (age, education etc). Variables such as age, education, course of disease, mental illness etc are just thrown in without going into any details. What do the authors mean by course of illness? Please also see my comment about reporting of p-values.

14. Table 3 is not clear. Headings are not proper.

15. The authors explanation regarding patients who have received treatment earlier having a higher MELASQOL score is not very clear. Patients who are more disturbed are more likely to seek treatment from more than one doctor (that is why they have received treatment earlier).

16. How does the QoL differ between countries? Between patients with diff skin types?

17. Was the study protocol registered at PROSPERO before the start of study?

---

## [Decision Letter · Decision Letter 1]

12 Oct 2021

PONE-D-21-20728R1Evaluating quality of life among melasma patients with the MELASQOL scale: a systematic reviewPLOS ONE

Dear Dr. Xiang,

Thank you for submitting your manuscript to PLOS ONE. After careful consideration, we feel that it has merit but does not fully meet PLOS ONE’s publication criteria as it currently stands. Therefore, we invite you to submit a revised version of the manuscript that addresses the points raised during the review process.

We look forward to receiving your revised manuscript.

Kind regards,

Feroze Kaliyadan, M.D.

Academic Editor

PLOS ONE

Journal Requirements:

Additional Editor Comments (if provided):

‘The relationship between the MELASQOL and MASI scores is found to be mixed’ – need to clarify in the abstract, what you mean by ‘mixed’

‘In addition, the MELASQOL score had no definite correlation with patient characteristics’ – like which characteristics?

For exclusion criteria, how did you decide on ‘patients who were unwilling or unable to understand the questionnaire’ and ‘incomplete and incorrect data’?

Will need to expand and briefly explain JBI

‘one study concluded that the relationship between the scores of MELASQOL and MASI were strongly correlated; the others’ results showed that although the two were correlated, the correlation was not strong’ What kind of correlation measure was used?

The section titled ‘Recommendations for practice and further research’ – it is not clear what the authors really want to convey here. The main focus should be on the gaps in knowledge unearthed in the systematic review and suggestions on how to address the same. While this is partially covered, a lot of the statements seem a bit vague and general

Reviewers' comments:

Reviewer's Responses to Questions

**Comments to the Author**

1. If the authors have adequately addressed your comments raised in a previous round of review and you feel that this manuscript is now acceptable for publication, you may indicate that here to bypass the “Comments to the Author” section, enter your conflict of interest statement in the “Confidential to Editor” section, and submit your "Accept" recommendation.

Reviewer #1: All comments have been addressed

Reviewer #2: (No Response)

2. Is the manuscript technically sound, and do the data support the conclusions?

Reviewer #1: Yes

Reviewer #2: Partly

3. Has the statistical analysis been performed appropriately and rigorously? 

Reviewer #1: I Don't Know

Reviewer #2: I Don't Know

4. Have the authors made all data underlying the findings in their manuscript fully available?

Reviewer #1: Yes

Reviewer #2: Yes

5. Is the manuscript presented in an intelligible fashion and written in standard English?

Reviewer #1: Yes

Reviewer #2: No

6. Review Comments to the Author

Reviewer #1: Most of the queries have been addressed.

Grammatic errors have creeped in. For eg; the term 'literatures' have been used in place of 'literature'

Reviewer #2: Dear Authors, Thank you for addressing the comments.

However, while some responses mentioned that additional information has now been added to the manuscript, this was not the case (eg domains of MELASQOL, Interpretability of MELASQOL, what is understood by 'course of disease' etc).

Some other questions have also emerged:

1. The inclusion/exclusion criteria for the purpose of a systematic review is relevant for studies only.

How is it relevant for participants? How would the authors exclude participants not meeting the criteria, as information for every participant is generally not available?

2. English language issues are still present. It may not be enough to be just grammatically correct, appropriate emphasis at important places should also be given. Overall, the manuscript is not easy to read.

3. The authors seem to have misinterpreted the point regarding 'physical health'. The patients feeling that melasma may be a manifestation of an underlying illness does not mean that they are physically unwell. This is just a patient perception, and would be seen as a problem in understanding of melasma or how they look at their melasma. In fact, there is no domain in MELASQOL which is related to physical health.

4. "We summarized and found no differences in quality of life between countries or patients with different skin types" Where is the supportive data in the manuscript? Table 1 shows a wide range of scores from different countries. How did you compare them? It may suffice that there is a variation in MELASQOL scores, with the lowest scores reported from... and highest from...

5. Table 1 - Why is the study by Yalamanchili included? You have not shown the MASI or MELASQOL scores from this study.

6. Table 3 - No mention of disease course; Not clear what is to be understood by disease course

Dogramaci et al - p<0.05, but mentioned under the column of statistically not significant

Misery et al - P value not given

Not clear what is older and younger patients; different studies might have used different age cut offs

A suggestion to authors - Instead of presenting the results study-wise, try presenting them for each patient variable (summarizing results of different studies for each variable) - Age, disease duration, previous treatment etc. That will give a clearer picture.

7. Fig 2 - How many patients' data available / used for this? Is it okay to create an average score like this? If so, what about the average of total MELASQOL score?

7. PLOS authors have the option to publish the peer review history of their article (what does this mean?). If published, this will include your full peer review and any attached files.

Reviewer #1: No

Reviewer #2: No

---

## [Editor Report · Decision Letter 2]

19 Nov 2021

PONE-D-21-20728R2Evaluating quality of life among melasma patients with the MELASQOL scale: a systematic review and meta-analysisPLOS ONE

Dear Dr. Xiang,

Thank you for submitting your manuscript to PLOS ONE. After careful consideration, we feel that it has merit but does not fully meet PLOS ONE’s publication criteria as it currently stands. Therefore, we invite you to submit a revised version of the manuscript that addresses the points raised during the review process.

We look forward to receiving your revised manuscript.

Kind regards,

Feroze Kaliyadan, M.D.

Academic Editor

PLOS ONE

Journal Requirements:

Additional Editor Comments:

The manuscript reads better overall after the second revision and the response to the reviewers' comments seem satisfactory, however the language still needs improvement. Please do a thorough recheck and revision for language and grammar.

The second area which still needs a bit of clarity is the inclusion/ exclusion criteria.

For the inclusion /exclusion criteria - you mention "For publication type: all article types (including original article, review, letter, etc.)" did you mean systematic reviews? This is not clear. Also, as I understand, you mention inclusion of an editorial. Can you please clarify regarding this.
---

## [Editor Report · Decision Letter 3]

6 Jan 2022

Evaluating the quality of life among melasma patients using the MELASQoL scale: a systematic review and meta-analysis

PONE-D-21-20728R3

Dear Dr. Xiang,

We’re pleased to inform you that your manuscript has been judged scientifically suitable for publication and will be formally accepted for publication once it meets all outstanding technical requirements.

Kind regards,

Feroze Kaliyadan, M.D.

Academic Editor

PLOS ONE
---

## [Editor Report · Acceptance letter]

10 Jan 2022

PONE-D-21-20728R3 

Evaluating the quality of life among melasma patients using the MELASQoL scale: a systematic review and meta-analysis 

Dear Dr. Xiang:

I'm pleased to inform you that your manuscript has been deemed suitable for publication in PLOS ONE. Congratulations! Your manuscript is now with our production department. 

Kind regards, 

on behalf of

Dr. Feroze Kaliyadan 

Academic Editor

PLOS ONE